# Evaluation of *Usnea barbata* (L.) Weber ex F.H. Wigg Extract in Canola Oil Loaded in Bioadhesive Oral Films for Potential Applications in Oral Cavity Infections and Malignancy

**DOI:** 10.3390/antiox11081601

**Published:** 2022-08-19

**Authors:** Violeta Popovici, Elena Matei, Georgeta Camelia Cozaru, Laura Bucur, Cerasela Elena Gîrd, Verginica Schröder, Emma Adriana Ozon, Oana Karampelas, Adina Magdalena Musuc, Irina Atkinson, Adriana Rusu, Simona Petrescu, Raul-Augustin Mitran, Mihai Anastasescu, Aureliana Caraiane, Dumitru Lupuliasa, Mariana Aschie, Victoria Badea

**Affiliations:** 1Department of Microbiology and Immunology, Faculty of Dental Medicine, Ovidius University of Constanta, 7 Ilarie Voronca Street, 900684 Constanta, Romania; 2Center for Research and Development of the Morphological and Genetic Studies of Malignant Pathology, Ovidius University of Constanta, CEDMOG, 145 Tomis Blvd., 900591 Constanta, Romania; 3Clinical Service of Pathology, Sf. Apostol Andrei Emergency County Hospital, 145 Tomis Blvd., 900591 Constanta, Romania; 4Department of Pharmacognosy, Faculty of Pharmacy, Ovidius University of Constanta, 6 Capitan Al. Serbanescu Street, 900001 Constanta, Romania; 5Department of Pharmacognosy, Phytochemistry, and Phytotherapy, Faculty of Pharmacy, Carol Davila University of Medicine and Pharmacy, 6 Traian Vuia Street, 020956 Bucharest, Romania; 6Department of Cellular and Molecular Biology, Faculty of Pharmacy, Ovidius University of Constanta, 6 Capitan Al. Serbanescu Street, 900001 Constanta, Romania; 7Department of Pharmaceutical Technology and Biopharmacy, Faculty of Pharmacy, Carol Davila University of Medicine and Pharmacy, 6 Traian Vuia Street, 020956 Bucharest, Romania; 8“Ilie Murgulescu” Institute of Physical Chemistry, Romanian Academy, 202 Spl. Independentei, 060021 Bucharest, Romania; 9Department of Oral Rehabilitation, Faculty of Dental Medicine, Ovidius University of Constanta, 7 Ilarie Voronca Street, 900684 Constanta, Romania

**Keywords:** *Usnea barbata* (L.) Weber ex F.H. Wigg, canola oil, bioadhesive oral films, poloxamer 407, antimicrobial activity, *Artemia salina*, blood cells, CLS-354 tumor cell line, anticancer activity

## Abstract

*Usnea* lichens are known for their beneficial pharmacological effects with potential applications in oral medicine. This study aims to investigate the extract of *Usnea barbata* (L.) Weber ex F.H. Wigg from the Călimani Mountains in canola oil as an oral pharmaceutical formulation. In the present work, bioadhesive oral films (F-UBO) with *U. barbata* extract in canola oil (UBO) were formulated, characterized, and evaluated, evidencing their pharmacological potential. The UBO-loaded films were analyzed using standard methods regarding physicochemical and pharmacotechnical characteristics to verify their suitability for topical administration on the oral mucosa. F-UBO suitability confirmation allowed for the investigation of antimicrobial and anticancer potential. The antimicrobial properties against *Staphylococcus aureus* ATCC 25923, *Pseudomonas aeruginosa* ATCC 27353, *Candida albicans* ATCC 10231, and *Candida parapsilosis* ATCC 22019 were evaluated by a resazurin-based 96-well plate microdilution method. The brine shrimp lethality assay (BSL assay) was the animal model cytotoxicity prescreen, followed by flow cytometry analyses on normal blood cells and oral epithelial squamous cell carcinoma CLS-354 cell line, determining cellular apoptosis, caspase-3/7 activity, nuclear condensation and lysosomal activity, oxidative stress, cell cycle, and cell proliferation. The results indicate that a UBO-loaded bioadhesive film’s weight is 63 ± 1.79 mg. It contains 315 µg UBO, has a pH = 6.97 ± 0.01, a disintegration time of 124 ± 3.67 s, and a bioadhesion time of 86 ± 4.12 min, being suitable for topical administration on the oral mucosa. F-UBO showed moderate dose-dependent inhibitory effects on the growth of both bacterial and fungal strains. Moreover, in CLS-354 tumor cells, F-UBO increased oxidative stress, diminished DNA synthesis, and induced cell cycle arrest in G0/G1. All these properties led to considering UBO-loaded bioadhesive oral films as a suitable phytotherapeutic formulation with potential application in oral infections and neoplasia.

## 1. Introduction

Plants have been used in various disease therapies since ancient times; numerous individual cultures use specific plants for medicinal purposes. Traditional medicine—according to the World Health Organization (WHO)—is a sum of knowledge, skills, and practices based on beliefs, theories, and experiences indigenous to different cultures, used for maintaining health and involved in physical and mental illness diagnosis, treatment, amelioration, and prevention [1]. In their study [2], Davidson and Frank pointed out the significant differences between the most common terms belonging to this domain. Therefore, the medicinal use of numerous plants has been named plant medicine, while phytomedicine defines the current application of plant-based medicine. Phytopharmaceuticals are plant-based standardized extracts manufactured by authorized pharmaceutical producers [2]. In Europe, phytomedicine represents a part of the mainstream medical practice, and in hospitals, it is used primarily as adjuvant therapy. In addition, numerous general practitioners prescribe phytopharmaceuticals, and most of these prescriptions are supported by the public health insurance system in various European countries [2].

As medicinal plants, lichens are a promising source of antibiotic and anticancer drugs [3]. Auerbach’s Wilderness Medicine Seventh Edition (2017) [2] includes the lichens from the genus *Usnea* (*Parmeliaceae*) in the North American Plant Medicines with therapeutical applications. *Usnea* sp. (old man’s beard) has been used since ancient times, having a broad use across many cultures throughout the world: as an antiseptic (Argentina), eupeptic (Italy), wound healer (Canary Islands), antibacterial (Saudi Arabia), and antitumor agent (Chile) [2,4]. In Europe, *Usnea* lichens have been commonly used as a topical medication due to usnic acid’s antibacterial, anti-inflammatory, and analgesic properties [5]. In addition to inhibitory effects on Gram-positive and Gram-negative bacteria, usnic acid has been shown to have other valuable activities (antiviral, antiprotozoal, and antiproliferative) consistent with its traditional use [6]. Soft and wispy *Usnea* sp. can be applied directly to the affected area and held in place by whatever means; dried lichen may be powdered and sprinkled directly on wounds, affording antimicrobial wound protection [2].

Usnic acid is the most studied lichen secondary metabolite and one of the few commercially available until recently. It was used as a dietary supplement for weight loss and associated with liver failure [7] and contact dermatitis [8,9]. For this reason, the current use is topical, including many different salves and creams with antimicrobial and anti-inflammatory action [10] and lozenges for oral cavity inflammation [2].

A recent literature review [11] also reported that studies on *Usnea* sp. bioactivities are limited compared to studies on its significant metabolite, usnic acid. The toxicity data examined by Sepahvand et al. (2021) evidenced that the use of pure usnic acid, mainly (−) enantiomer, is associated with the hepatotoxic effect. However, in most *Usnea* sp., usnic acid is identified as (+) enantiomer and based on their information, the extracts of *Usnea* sp. can be considered safer products [11].

In their study, Prateeksha et al. proved that *Usnea* lichens have valuable pharmacological potential and evidenced them as a potent phytomedicine [12]. Numerous studies investigated *Usnea* sp. extracts and displayed their antioxidant [13,14,15], antimicrobial [16,17,18], and anticancer effects [19,20,21]. Most authors have used chemical solvents for the preparation of these lichen extracts: ethanol, methanol, acetone, ethyl acetate, dimethyl ether, CCl_4_, hexane, and CO_2_ supercritical [22,23,24]. We propose to explore the properties of *U. barbata* extract in a green solvent—canola oil [25,26].

A recently published article [27] reported an oral product formulated as compressed tablets based on plant extracts/essential oils containing *U. barbata* supercritical CO_2_ extract. This work investigates a different pharmaceutical formulation using *U. barbata* extract in canola oil, with potential application in oral medicine.

An interesting aspect for dental professionals is the presence of poloxamer 407 (P407) as an emulsifier for oil extract. Poloxamer 407 is a nonionic surfactant used in mouthwashes and toothpaste belonging to the most known brands: Listerine (Johnson & Johnson Healthcare Products Division of McNEIL-PPC, Inc., Fort Washington, PA, USA) and Colgate (Colgate-Palmolive Company, New York, NY, USA) [28] and other cosmetic products. P407 bioactivities were examined in our work, this compound being selected as a positive control for all performed studies.

The present study aimed to formulate and develop bioadhesive oral films loaded with canola oil extract of *U. barbata* (L.) F. H. Wigg and evaluate their antibacterial and antifungal properties. It also explored the F-UBO cytotoxicity on an animal model and in vitro anticancer activity.

## 2. Materials and Methods

### 2.1. Materials

This study’s chemicals, reagents, and standards were of analytical grade. Usnic acid standard 98.1% purity, propidium iodide (PI) 1.0 mg/mL, dimethyl sulfoxide (DMSO), Poly (ethylene glycol)-block-poly (propylene glycol)-block-poly (ethylene glycol) (poloxamer 407), polyethylene glycol 400 (PEG 400), and hydroxypropyl methylcellulose (HPMC) and antibiotics mix solution—100 µL/mL with 10 mg streptomycin, 10,000 U penicillin, 25 µg amphotericin B per 1 mL—were provided by Sigma-Aldrich Chemie GmbH (Taufkirchen, Germany). Annexin V Apoptosis Detection Kit and flow cytometry staining buffer (FCB) were purchased from eBioscience^TM^ (Frankfurt am Main, Germany) and RNase A 4 mg/mL from Promega Corporation (Madison, WI, USA). Magic Red^®^ Caspase-3/7 Assay Kit, Reactive Oxygen Species (ROS) Detection Assay Kit, and EdU i-Fluor 488 Kit were supplied by Abcam (Cambridge, UK).

The OSCC cell line (CLS-354) growing culture and the culture medium—Dulbecco’s Modified Eagle’s Medium (DMEM) high glucose with 10% FBS, basic supplemented with 4.5 g/L glucose, 2 mM L-glutamine, and 10% fetal bovine serum (FBS)—were provided by CLS Cell Lines Service GmbH (Eppelheim, Germany). Trypsin-ethylenediamine tetra acetic acid (Trypsin EDTA) and the media for blood cells—Dulbecco’s phosphate-buffered saline with MgCl_2_ and CaCl_2_, fetal bovine serum (FBS) and L-glutamine (200 mM) solution—were purchased from Gibco^TM^ Inc. (Billings, MT, USA).

The blood samples were collected from a non-smoker healthy donor (B Rh+ blood type), according to Ovidius University of Constanta Ethical approval code 7080/10.06.2021 and Donor Consent code 39/30.06.2021.

*U. barbata* lichen was harvested in March 2021 from the branches of conifers in the forest localized in the Călimani Mountains (47°29′ N, 25°12′ E, and 900 m altitude). It was identified by the Department of Pharmaceutical Botany of the Faculty of Pharmacy, Ovidius University of Constanta, using standard methods. A voucher specimen is maintained in the Herbarium of Pharmacognosy Department, Faculty of Pharmacy, Ovidius University of Constanta (Popovici 3/2021, Ph-UOC). The canola seed oil for oil extract preparation was provided by TAF PRESSOIL SRL, Cluj, Romania.

*Artemia salina* eggs and Artemia salt (Dohse Aquaristik GmbH & Co., Gelsdorf, Germany) were purchased online from https://www.aquaristikshop.com/ (accessed on 5 May 2022).

The microbial cell lines (*S. aureus* ATCC 25923, *P. aeruginosa* ATCC 27353, *C. albicans* ATCC 10231, and *C. parapsilosis* ATCC 22019) were obtained from the Microbiology Department, S.C. Synevo Romania SRL, Constanta Laboratory, in partnership agreement No 1060/25.01.2018 with the Faculty of Pharmacy, Ovidius University of Constanta. Culture medium Mueller-Hinton agar (MHA) was supplied by Thermo Fisher Scientific, GmbH, Dreieich, Germany; RPMI 1640 medium and resazurin solution (from In Vitro Toxicology Assay Kit, TOX8-1KT, resazurin-based) were purchased from Sigma-Aldrich Chemie GmbH (Taufkirchen, Germany).

### 2.2. Formulation and Preparation of the UBO-Loaded Bioadhesive Oral Films

The *U. barbata* extract in canola oil (UBO) was obtained using a method adapted from that described by Basiouni et al. [29], from 20.2235 g dried and ground lichen and 500 mL cold-pressed canola seed oil [26] in darkness, at room temperature (21–22 °C). The container with both components was shaken daily for three months; then, UBO was filtered in another brown vessel with a sealed plug and preserved in a plant room, sheltered from sun rays. Both oil samples had a pH of 4.

For the development of the bioadhesive oral films containing *U. barbata* extract in canola oil (F-UBO), HPMC K100 (with a viscosity of 100 mPa) was selected as the film-forming polymer. It displays excellent hydrophilicity, water-absorbing ability, good biocompatibility, and biodegradability [30,31,32]. PEG 400 was included in the films’ formulations as an external plasticizer for its high hydrophilic character and non-toxicity [33].

The non-ionic surfactant poloxamer 407 (P407) was chosen for emulsifying the oily phase to ensure the uniform incorporation of UBO in the polymer matrix.

Bioadhesive films, containing suitable excipients but no active ingredient load, were prepared and used as a reference (R) to prove the UBO activity and influence on the F-UBO pharmaceutical characteristics.

The formulation of the developed F-UBO is presented in Table 1.

The UBO amount was selected according to the emulsifying ability of the poloxamer 407 (P407) and film-forming polymer. HPMC was weighed using a Mettler Toledo AT261 balance (Marshall Scientific, Hampton, NH, USA) with 0.01 mg sensitivity for the polymeric matrix system. Then, it was dispersed in water by stirring at 700 rpm and room temperature, using an MR 3001K magnetic stirrer (Heidolph Instruments GmbH & Co. K.G., Schwabach, Germany) and mixed with PEG 400. P407 was dissolved in the matrix, and UBO was included under continuous stirring for 1 h.

References (R) were realized by mixing P407 with the base system, and the formed gels were left overnight at room temperature for deaeration. The viscous dispersions were poured in a thin layer into Petri glass plates and dried in ambient conditions for 24 h. Finally, the dried films were peeled off the plate surface and cut into 1.5 × 2 cm patches.

The manufacturing process led to defining the concentration of UBO in the film formulation: F-UBO encloses 315 µg UBO.

### 2.3. Physico-Chemical Analysis of Bioadhesive Oral Films

#### 2.3.1. SEM Analysis

A scanning electron microscope (SEM) in a high-resolution Quanta3D FEG (Thermo Fisher Scientific, GmbH, Dreieich, Germany) was used to investigate the bioadhesive oral film morphology.

#### 2.3.2. Atomic Force Microscopy (AFM)

The bioadhesive oral film’s morphology was obtained via atomic force microscopy (AFM). The AFM measurements were registered with an AFM XE-100 (Park Systems Corporate, Suwon, Korea) assisted with flexure-guided, crosstalk eliminated scanners in non-contact mode to minimize the tip–sample interaction. AFM images were registered with sharp tips (PPP-NCLR, from NANOSENSORS™, Neuchatel, Switzerland) having the following characteristics: less than 10 nm radius of curvature, 225 mm mean length, 38 mm mean width, ~48 N/m force constant, and a resonance frequency of 190 kHz. An XEI program (v 1.8.0—Park Systems Corporate, Suwon, Korea) was carried out to process the AFM images and to evaluate the roughness. The surface profile of the scanned samples (the dimensions of the selected particles indicated with red arrows along the selected line) shows the representative line scans presented below the AFM images in the so-called “enhanced contrast” mode.

#### 2.3.3. FTIR Analysis

The infrared spectra of the materials (obtained as bioadhesive films) were registered using FTIR equipment (Nicolet Spectrometer 6700 FTIR from Thermo Electron Corporation, Waltham, MA, USA) assisted with a diamond-crystal ATR accessory. In transmittance mode, data were acquired in the spectral range of 400–4000 cm^−1^ (resolution of 4 cm^−1^ and a total of 32 scans per spectrum).

#### 2.3.4. X-ray Diffraction Patterns

A Rigaku Ultima IV diffractometer (Rigaku Corporation, Tokyo, Japan) in parallel beam geometry was used to investigate the X-ray diffraction (XRD) patterns of bioadhesive films. A step size of 0.02 and a 2° (2θ)/min speed over 10–80° were used. The source of the X-rays was a CuKα tube (λ = 1.54056 Å) operating at 40 kV and 30 mA.

#### 2.3.5. Thermogravimetric Analysis

The thermal investigations were conducted on a Mettler Toledo TGA/SDTA851e thermogravimetric analyzer (Mettler-Toledo GmbH, Greifensee, Switzerland). The nonisothermal measurements were performed in the 25–600 °C temperature range, under 80 mL min^−1^ synthetic air atmosphere at a constant heating rate of 10 °C min^−1^.

### 2.4. Pharmacotechnical Analysis of Bioadhesive Oral Films

#### 2.4.1. Weight Uniformity

The weight uniformity was evaluated on 20 bioadhesive oral films of each formulation (F-UBO and R). They were individually weighed, and the average weight was determined.

#### 2.4.2. Thickness

This parameter was also measured on 20 bioadhesive films of each type (F-UBO and R) using a Yato digital micrometer (Yato China Trading Co., Ltd., Shanghai, China) with a measuring range of 0–25 mm and a resolution of 0.001 mm. Then, the mean value was calculated.

#### 2.4.3. Folding Endurance

The bioadhesive films were repeatedly folded and rolled until they broke, or up to 300 times [5]. The folding times were registered and expressed as folding endurance values.

#### 2.4.4. Tensile Strength and Elongation Ability

The film’s tensile strength and elongation ability were determined using an L.R. 10K Plus digital tensile force tester for universal materials (Lloyd Instruments Ltd., West Sussex, United Kingdom). The analysis was performed from a 30 mm distance with a speed of 30 mm/min. Therefore, each film was placed in a vertical position between the two braces, and the breakage force was registered. The measurements were achieved in triplicate.

The following equations (Equations (1) and (2)) were used to calculate the tensile strength and the elongation at break:(1)$$\mathrm{Tensile}\;\mathrm{strength}\;\left(\mathrm{kg}/{\mathrm{mm}}^{2}\right)=\frac{\mathrm{Force}\;\mathrm{at}\;\mathrm{breakage}\;\left(\mathrm{kg}\right)}{\mathrm{Film}\;\mathrm{thickness}\;\left(\mathrm{mm}\right)-\mathrm{Film}\;\mathrm{width}\;\left(\mathrm{mm}\right)}$$
(2)$$\mathrm{Elongation}\;\%\;=\frac{\mathrm{Increase}\;\mathrm{in}\;\mathrm{film}\;\mathrm{length}}{\mathrm{Initial}\;\mathrm{film}\;\mathrm{length}}\times 100$$

#### 2.4.5. Moisture Content

The moisture content was assessed as the loss on drying by the thermogravimetric method using an H.R. 73 halogen humidity analyzer (Mettler-Toledo GmbH, Greifensee, Switzerland) [6]. Five bioadhesive oral films of each formulation (F-UBO and R) were analyzed for moisture content determination.

#### 2.4.6. Surface pH

Five films of each formulation (F-UBO and R) were moistened with 1 mL of distilled water (pH 6.5 ± 0.5) for 5 min at room temperature. Then, the pH value was measured with the CONSORT P601 pH-meter (Consort bvba, Turnhout, Belgium).

#### 2.4.7. In Vitro Disintegration Time

The time required to disintegrate the F-UBO and R bioadhesive oral films, with no residual mass completely, was measured in simulated saliva phosphate buffer pH of 6.8 at 37 ± 2 °C, using an Erweka DT 3 apparatus (Erweka^®^ GmbH, Langen, Germany) [34].

#### 2.4.8. Swelling Rate

Six films of each formulation (F-UBO and R) were placed on 1.5% agar gel in Petri plates and incubated at 37 ± 1 °C. Every 30 min, for 6 h, the patches were weighed. The swelling rate was calculated according to Equation (3):(3)$$\mathrm{Swelling}\;\mathrm{rate}\;=\frac{\mathrm{Wt}-\mathrm{Wi}}{\mathrm{Wi}}\times 100$$
where w_t_ is the patch weight at time t after the incubation and w_i_ is the initial weight [35,36,37,38].

#### 2.4.9. Ex Vivo Bioadhesion Time

The bioadhesion time [39] was measured through the method described by Gupta et al. [40] on a detached porcine buccal mucosa by removing the fat layer and any residual tissue. The buccal mucosa was washed with distilled water and a phosphate buffer (pH 6.8) at 37 °C and fixed on a glass plate. Each bioadhesive film was hydrated in the center with 15 μL phosphate buffer and brought on the mucosa surface by pressing it for 30 s. The glass plate was placed in 200 mL phosphate buffer pH 6.8 and maintained at 37 °C for 2 min. The suitable simulation of the oral cavity conditions was ensured using a paddle with a stirring rate of 28 rpm. Then, the necessary time for the entire film’s erosion or detachment from the buccal mucosa surface was recorded [41]. This registered time represents the film’s residence time [42] on the oral mucosa, known as a bioadhesion time [43]. This test was realized in triplicate.

### 2.5. F-UBO Antimicrobial Activity Evaluation by Resazurin-Based 96-Well Plate Microdilution Method

#### 2.5.1. Inoculum Preparation

The bacterial inoculum was prepared by the direct colony suspension method [44]. Thus, bacterial colonies selected from a 24 h agar plate were suspended in M.H.A. medium, according to the 0.5 McFarland standard, measured at Densimat Densitometer (Biomerieux, Marcy-l’Étoile, France) with around 108 CFU/mL (CFU = colony-forming unit), The yeast inoculum was prepared using the same method, adjusting the RPMI 1640 with fungal colonies to the 1.0 McFarland standard, with 10^6^ CFU/mL.

#### 2.5.2. Samples and Standards

F-UBO was dissolved in 1 mL of diluted phosphate buffer. As standards, ceftriaxone (Cefort 1g Antibiotice SA, Iasi, Romania) solutions 30 mg/mL and 122 mg/mL in distilled water were used for bacteria. The Cefort powder was weighted at Partner Analytical balance (Fink & Partner GmbH, Goch, Germany) and dissolved in distilled water. Terbinafine solution 10.1 mg/mL (Rompharm Company SRL, Otopeni, Romania) was used as standard for *Candida* sp. As a positive control for antimicrobial activity evaluation, 5% P407 was selected, the emulsifier used for the UBO-loaded bioadhesive oral films formulation.

#### 2.5.3. Microdilution Method

All successive steps were performed in an Aslair Vertical 700, laminar flow, microbiological protection cabinet (Asal Srl, Cernusco (MI), Italy). In four 96-well plates, we performed seven serial dilutions, adapting the protocol described by Fathi et al. [45].

All 96-well plates were incubated for 24 h at 37 °C for bacteria and 35 °C for yeasts in a My Temp mini Z763322 Digital Incubator (Benchmark Scientific Inc., Sayreville, NJ, USA).

#### 2.5.4. Reading and Interpreting

After 24 h incubation, the plates were examined with a free eye to see the color differences between standard and samples [46]. The corresponding sample concentration activities were compared with the Standard antibiotic ones. For yeasts, the color chart of the resazurin dye reduction method was used [47,48].

### 2.6. Evaluation of UBO-Loaded Bioadhesive Oral Films Cytotoxicity on Animal Model

Aiming to evaluate the F-UBO cytotoxicity, we used *Artemia salina* as an animal model, adapting a previously described method [49].

F-UBO film was placed in a diluted buffer (1 mL) and incubated for 15 min at 37 °C, resulting in a homogenous dispersion.

#### 2.6.1. Brine Shrimp Lethality Assay 

The larvae were obtained from *A. salina* cysts through continuous light and aeration in a 0.35% saline solution at 20 °C. The brine shrimp larvae in the first stage (instar I) were introduced in 0.3% saline solution into experimental pots (with a volume of 1 mL) [26]. The analysis was compared to a blank (untreated nauplii) to obtain accurate results regarding the F-UBO cytotoxic effect. An amount of 5% P407 in water was a positive control. The nauplii have embryonic energy reserves as lipids, and they were not fed during the test, thus avoiding interference with the sample and positive control. Their evolution was analyzed after 24 h and 48 h, exploring the morphological changes induced by F-UBO and P407 [50,51].

#### 2.6.2. Fluorescent Microscopy

The brine shrimp larvae were stained with 3% acridine orange (Merck Millipore, Burlington, MA, USA) for 5 min. The samples were subjected to drying for 15 min in darkness and placed on the microscope slides.

#### 2.6.3. Data Processing

The microscopic images were realized using a VWR microscope VisiScope 300D (VWR International, Radnor, PA, USA) with a Visicam X3 camera (VWR International Radnor, PA, USA) at 40×, 100×, and 400× magnification. They were processed with VisiCam Image Analyzer 2.13.

Fluorescent microscopy images were achieved using an OPTIKA B-350 microscope (Ponteranica, BG, Italy) blue filter (λex = 450–490 nm; λem = 515–520 nm) and green filter (λex = 510–550 nm; λem = 590 nm). The FM images at 100× and 400× magnification were processed with Optikam Pro 3 Software (OPTIKA SRL, Ponteranica, BG, Italy).

All observations were performed in triplicate.

### 2.7. In Vitro Cytotoxicity of UBO-Loaded Bioadhesive Oral Films on Human Normal Blood Cells and CLS-354 Tumor Cells

#### 2.7.1. Equipment

The present study platform for in vitro F-UBO cytotoxicity analysis was the Attune Acoustic focusing cytometer (Applied Biosystems, Bedford, MA, USA). Before cell analysis, the flow cytometer was first set by fluorescent beads—Attune performance tracking beads, labeling, and detection (Life Technologies, Europe BV, Bleiswijk, The Netherlands), with standard size (four intensity levels of beads population). The cell amount was established by counting cells below 1 µm [52]. Using forward scatter (FSC) and side scatter (SSC), more than 10,000 cells per sample for each analysis were gated.

#### 2.7.2. Data Processing

Flow cytometry data were processed using Attune Cytometric Software v.1.2.5, Applied Biosystems, 2010 (Bedford, MA, USA).

#### 2.7.3. Human Blood Cell Cultures

The blood samples were collected into heparin vacutainers, and the blood cell cultures were obtained according to a previously described method [53]. Then, the blood cells were treated with F-UBO and controls in Nunclon Vita Cell culture 6-well plates (Kisker Biotech GmbH & Co.KG, Steinfurt, Germany) and incubated in a Steri-Cycle™ i160 CO_2_ Incubator (Thermo Fisher Scientific Inc., Waltham, MA, USA), at 37 °C, in 5% CO_2_ for 24 h. All flow cytometry analyses were performed after this incubation time.

#### 2.7.4. CLS-354 Cell Line

The human mouth squamous cell carcinoma cell line CLS-354 (CLS catalog number 300152) consists of epithelial cells established in vitro from the primary squamous carcinoma of a 51-year-old Caucasian male. The CLS-354 cells [54] were cultured in DMEM high glucose with 10% FBS, supplemented with antibiotic mix solution in humidity conditions of 5% CO_2_ at 37 °C for 7 days. The cells were dissociated from the monolayer with trypsin–EDTA, centrifugated at 3000 rpm for 10 min in a Fisher Scientific GT1 centrifuge (Thermo Fisher Scientific Inc., Waltham, MA, USA), and distributed in Millicell™ 24-well cell culture microplates (Thermo Fisher Scientific Inc., Waltham, MA, USA). After treatment, they were incubated for 24 h in the same conditions. All the flow cytometry analyses were performed after this incubation period.

#### 2.7.5. Samples and Control Solutions

F-UBOs were dissolved in the suitable culture media for both types of cells, with 1% DMSO. As positive controls, 5% P407 and usnic acid of 125 ug/mL in 1% DMSO were used, and as a negative control, 1% DMSO.

#### 2.7.6. Annexin V-FITC Apoptosis Assay

The cells with annexin V-FITC and PI (20 µg/mL) were incubated in darkness, for 30 min, at room temperature [53]. Then, the viable, early apoptotic, late apoptotic, and necrotic cells were examined at a flow cytometer using a 488 nm excitation, green emission for annexin V-FITC (BL1 channel), and orange emission for PI (BL2 channel).

#### 2.7.7. Evaluation of Caspase-3/7 Activity

The cells were well-mixed with Magic Red^®^ Caspase-3/7 substrate solution and PI and incubated [53]. Then, the early stages of cell apoptosis by activating caspases-3/7 (DEVD-ases [55]) were analyzed through flow cytometry using a 488 nm excitation, red emission for MR-(DEVD)_2_ (BL3 channel), and orange emission for PI (BL2 channel).

#### 2.7.8. Evaluation of Nuclear Condensation and Lysosomal Activity

The cells were stained successively with Hoechst 33,342 and AO and incubated for 30 min at room temperature in darkness [53]. Then, they were examined at the flow cytometer, using UV excitation and blue emission for Hoechst 33,342 (VL2) at 488 nm and green emission for acridine orange (BL1 channel).

#### 2.7.9. Cell Cycle Analysis

The cells with PI (20 µg/mL) and RNase A (30 µg/mL) were incubated at room temperature, into darkness, for 30 min [53]. Next, the cell cycle distribution was detected by flow cytometry, using a 488 nm excitation and orange emission for PI (BL2 channel) [56].

#### 2.7.10. Evaluation of Total ROS Activity

ROS Assay Stain solution was well-mixed with cell cultures and incubated at 37 °C for 60 min [53]. Then, the cells were analyzed by flow cytometry, using a 488 nm excitation and green emission for ROS (BL1 channel).

#### 2.7.11. Evaluation of Cell Proliferation

The cell cultures were incubated for 2 h with 50 µM EdU (500 µL) at 37 °C. Following a succession of previously described steps [53], they were prepared for flow cytometry examination at a 488 nm excitation and green emission for EdU-iFluor 488 (BL1).

### 2.8. Data Analysis

All analyses were effectuated in triplicate, and the data were registered as means values ± standard deviation (SD). The results are expressed as percent (%) in the case of cell apoptosis, caspase-3/7 activity, nuclear condensation, autophagy, cell cycle arrest, and DNA synthesis, and count (×10^4^) for ROS levels. Data analysis was realized with SPSS v. 23 software, IBM, 2015. Paired *t*-test established the differences between F-UBO and controls, and *p* < 0.05 was considered statistically significant. The principal component analysis was performed with XLSTAT 2022.2.1. by Addinsoft (New York, NY, USA) and examined the correlations between variable parameters.

## 3. Results

### 3.1. Organoleptic Characteristics of Bioadhesive Oral Films

The organoleptic characteristics of the F-UBO and R films depend highly on the active ingredient state. Both bioadhesive films (R and F-UBO) are white, with the typical appearance of emulsified systems (Figure 1a,b). The films withstand normal handling and cutting processes without air bubbles, cracks, or imperfections. All formulations lead to homogenous, thin, and easy-to-peel bioadhesive oral films, with a uniform, smooth, and glossy surface (Figure 1a,b).

### 3.2. Physico-Chemical Characterization of Bioadhesive Oral Films

#### 3.2.1. Morphology

Scanning electron microscopy (SEM) was performed to study the morphology of the bioadhesive films (Figure 1c,d). SEM image of R (Figure 1c) shows a denser surface containing few cavities and small spherically shaped protrusions. The F-UBO surface observed by SEM analysis (Figure 1d) is rough with deeper and interconnected pores compared to R.

#### 3.2.2. Atomic Force Microscopy

The AFM images are displayed in Figure 1e,f.

The reference (R) is corrugated, with a surface exhibiting large protruding particles (ranging from tens of nm up to microns—for example, see the upper, middle-left elongated particle in Figure 1e). Therefore, R is characterized by a higher global RMS roughness of 8.5 nm and an Rpv parameter of 804.5 nm. The line scan exhibits a vertical gradient (Δz level difference) of ~180 nm and more prominent surface features (Figure 1e).

F-UBO displays a rougher surface, having an RMS roughness of 98.5 nm and a peak-to-valley parameter of 480.6 nm (Figure 1f). The compact morphology is maintained similar to that of R; however, deep grooves and cavities are seen in the image, such as the one imaged along the red line, which is more than 300 nm deep. The surface features (such as pits, cavities, and grooves) create a clear surface corrugation, which could enhance the bioadhesive films’ adherence to the targeted tissue.

#### 3.2.3. FTIR Spectra

The FTIR spectra of R and F-UBO are illustrated in Figure 2.

The literature data showed that the main absorption peaks characterize the FTIR spectrum of P407 at 2893 cm^–1^ due to C-H stretch aliphatic, 1355 cm^–1^ corresponding to in-plane O-H bend, and 1124 cm^–1^ due to C-O stretch [57]. In addition, the FTIR spectrum of pure HPMC shows an absorption band at 3444 cm^−1^ assigned to the stretching frequency of the hydroxyl (-O.H.) group. The band at 1373 cm^−1^ is due to bending vibration of -O.H. Other stretching vibration bands related to C-H and C-O can be noted at 2929 cm^−1^ and 1055 cm^−1^, respectively (Figure 2).

The prominent peaks of P407 and pure HPMC were shifted in R and F-UBO films due to the formation of bioadhesive film [58,59]. The main FTIR peaks of UBO [60] are superposed to the peaks of the polymer matrix.

On the other hand, the spectra exhibit the νO-H stretching vibration detected at 3460 cm^−1^ and νsim CH_2_ at 2853 cm^−1^, characteristic of P407. The band observed at 1050 cm^−1^ was assigned to the C-O group [61] (Figure 2). These findings are in accord with the assumption that F-UBO bioadhesive films are formed through UBO dispersion in the polymer matrix.

#### 3.2.4. X-ray Diffractograms

The X-ray diffractograms of bioadhesive oral films are presented in Figure 3a.

The X-ray diffractograms of R and F-UBO exhibited two peaks at 2θ = 8° and 2θ = 20°. They represent the prominent HPMC XRD peaks, portraying their semicrystalline structure [62,63]. The peak at 2θ = 22° is attributed to P407 [64]. The X-ray diffraction pattern of F-UBO shows higher intensity peaks than the reference, proving the influence of UBO dispersion in the polymer matrix and correlated with the bioadhesive behavior.

#### 3.2.5. Thermogravimetric Analysis

Thermogravimetric analysis coupled with differential thermal analysis was performed to characterize the film’s thermal behavior and stability. Both materials (reference and F-UBO films) exhibit a similar behavior upon heating from 25 to 600 °C (Figure 3b). A 0.8–2.5% weight mass loss occurs on heating up to ~100 °C, which can be associated with the loss of residual solvent and physisorbed water. The decomposition process of the organic compounds takes place in two distinct steps, between 200–400 °C and 400–550 °C. Each decomposition step is accompanied by an exothermic thermal effect (Figure 3b). The mass losses associated with the solvent loss and first and second organic decomposition steps are presented in Table 2.

Figure 3b and Table 2 indicate that the first stage of both bioadhesive films (R and F-UBO) starts at a temperature below 100 °C due to the loss of adsorbed water. The second stage begins from 230 °C to 380 °C, corresponding to ~86% (for R) and ~87% (for F-UBO) weight loss. The third stage, with a maximum of 495.3 °C (for R) and 488.8 °C (for F-UBO) was due to the different organic part decomposition.

### 3.3. Pharmacotechnical Evaluation of Bioadhesive Oral Films

The results of pharmacotechnical evaluation of F-UBO and R are presented in Table 3. The film’s weight varies depending on the active ingredient state and dispersion method. No significant differences are registered between the UBO-loaded films and the reference films (63 ± 1.79 vs. 62 ± 3.27 mg, *p* > 0.05).

The thickness of F-UBO is similar to that of R (0.069 ± 0.006 vs. 0.065 ± 0.004 mm, *p* > 0.05). Low SD values registered in thickness measurements (Table 3) could also be observed.

Both films displayed a great folding endurance, with values above 300, proving suitable flexibility. The UBO-loaded films show a higher elongation and a lower tensile strength than the references (56.33 ± 0.92 vs. 52.16 ± 1.22, *p* < 0.05; 2.17 ± 0.49 vs. 2.36 ± 0.98, *p* > 0.05), proving the active ingredient’s influence on the film’s resistance and elasticity. The F-UBO’s moisture content reported minor differences compared to that of R (8.11 ± 0.78 vs. 8.42 ± 0.69, *p* > 0.05). In addition, the pH measured on the film’s surface shows approximately neutral and similar values for both formulations (6.97 ± 0.01 vs. 7.02 ± 0.04, *p* > 0.05). Furthermore, another three pharmacotechnical properties do not show significant differences between F-UBO and R: in vitro disintegration time in a simulated saliva medium (124 ± 3.67 vs. 127 ± 4.81, *p* > 0.05), swelling rate (195 ± 5.24 vs. 202 ± 5.68, *p* > 0.05), and bioadhesion time (86 ± 4.12 vs. 91 ± 3.79, *p* > 0.05).

The swelling rate over 6 h is presented in Figure 4.

Figure 4 indicates that the swelling index increases linearly in the first 4 h (approximately 20% to every 30 min); then, the growth becomes slower, the differences between 330 and 360 min being insignificant. No films were eroded after 6 h, and no swelling could be detected after this time. With the oily active ingredient emulsified in the matrix, F-UBO displays a lower swelling behavior than R due to UBO’s state and dispersion (Table 3). 

### 3.4. Antimicrobial Activity

Data registered in Table 4 show the standard antibiotic (CTR), antifungal drug (TRF), P407, and F-UBO initial concentrations and microdilutions (mg/mL).

The results obtained after 24 h incubation at 37 °C are displayed in Appendix A.

Data from Appendix A indicate that the color showed by the standard antibiotic correlates with its inhibiting power and varies in a manner directly proportional to its concentration. CTR induced a “moderate” to “good” inhibition of bacterial strains’ growth; the microdilutions [67,68] from 30.230 mg/mL CTR were less active than those from 122.330 mg/mL.

F-UBO exhibited inhibitory effects on both bacteria tested (Appendix A). Comparing the colors of the well-plates with F-UBO and CTR, it can be observed that F-UBO of [3.176–0.100] mg/mL acts against *S. aureus*, similar to CTR of [0.047–0.023] mg/mL. F-UBO of [3.176—0.199] mg/mL inhibits *P. aeruginosa* proliferation, similar to CTR of [0.047–0.023] mg/mL (Appendix A).

In the present study, 5% P407 had inhibitory effects against both bacteria tested (Appendix A), and its action was higher against *P. aeruginosa* than *S. aureus.* P407 of [2.506–0.626] mg/mL acted on *P. aeruginosa*, similar to CTR of [1.511—0.047] mg/mL. Moreover, P407 of [2.506–1.253] mg/mL inhibited *S. aureus* proliferation, similar to CTR of [0.047–0.023] mg/mL. Compared to F-UBO, 5% P407 reported similar effects on *S. aureus* and induced a slowly higher inhibition of *P. aeruginosa.*

Appendix A shows that terbinafine of [0.500–0.007] mg/mL exhibited the highest antifungal activity, having a fungicidal effect on both *Candida* sp.

F-UBO inhibited both *Candida* sp. proliferation. Thus, F-UBO of 3.176 mg/mL had the most significant inhibitory activity on both species, higher on *C. albicans* than on *C. parapsilosis.* The following decreasing F-UBO concentrations moderately inhibited the *Candida* sp. proliferation.

The antifungal activity of P407 is displayed in Appendix A. P407 of [2.506—0.078] mg/mL had a significant inhibitory effect, partially inducing the death of both fungal species [48]. The lowest concentration (0.039 mg/mL) similarly affected *C. albicans* and produced a moderate to fast proliferation of *C. parapsilosis* [48]. Moreover, Appendix A show that 5% P407 had considerably higher inhibitory activity on both *Candida* sp. than F-UBO.

### 3.5. Evaluation of UBO-Loaded Bioadhesive Oral Films Cytotoxicity on Animal Model

*A. salina* nauplii were examined under the microscope to detect morphological changes after 24 and 48 h of exposure, compared to a blank and positive control (5% P407). All these data are illustrated in Figure 5.

After 24 h, all larvae were alive, swimming, and showing normally visible movements. However, F-UBO cytotoxicity was revealed after the first 24 h, even if the larvae were alive and apparently normal. Compared to untreated larvae (Figure 5a–d), the images 400× (Figure 5e–h) show the following processes in the early stage: penetration of emulsified lipids into tissues, depletion of cellular structures, and minimal detachment of the cuticle from the terminal portion of the digestive tract.

These morphological changes were intensified over the next 24 h, becoming incompatible with the brine shrimp nauplii survival. Hence, after 48 h, 64.81% of larvae were alive, and 11.11% were in the sublethal stage; the registered mortality was 25.92%.

Compared to blank (Figure 5i–l), the exposed larvae (Figure 5m–p) had blocked digestive transit due to accumulated lipids, cell damage with large intercellular spaces, tissue destruction, and massive detachment of the cuticle from larval tissues.

The *A. salina* larvae were alive and had normal movements after 24 and 48 h exposure at 5% P407. Microscopic examination after 48 h revealed a significant digestive tube volume increase, especially in the superior part, and low penetration of emulsified lipids into tissue (Figure 5v–x).

Moreover, at the intracellular level, FM images (Figure 5A–F) show activated lysosomes in cell death processes in brine shrimp larvae exposed to F-UBO (Figure 5F).

### 3.6. In Vitro Cytotoxicity of UBO-Loaded Bioadhesive Oral Films on Human Normal Blood Cells and CLS-354 Tumor Cells

#### 3.6.1. Annexin V-FITC Apoptosis Assay

The effects of F-UBO on normal blood cells and CLS-354 tumor cells based on morphology and cell membrane integrity are illustrated in Figure 6.

After 24 h treatment with F-UBO, the normal blood cell’s viability (V) was significantly higher compared to the C3UA positive control: 85.43 ± 2.01 vs. 61.43 ± 0.88, *p* < 0.01 (Figure 6A,E,I).

The viability of CLS-354 tumor cells treated with F-UBO was also appreciably increased compared to both positive controls: 99.50 ± 0.72 vs. C2P: 72.51 ± 2.51; C3UA: 54.05 ± 1.68, *p* < 0.01 (Figure 6B,G,H,J).

Moreover, Figure 6 shows that F-UBO did not induce early apoptosis in both cell types, thus reporting considerable differences compared to C3UA in normal blood cells (0.00 ± 0.00 vs. 37.04 ± 0.66, *p* < 0.01), and both positive controls in CLS-354 tumor cells (0.00 ± 0.00 vs. C2P: 5.88 ± 1.24, *p* < 0.05 and C3UA: 12.92 ± 1.35, *p* < 0.01).

#### 3.6.2. Evaluation of Caspase-3/7 Activity

The pro-apoptotic signal induced by F-UBO in normal blood cells and CLS-354 tumor cells was achieved by measuring the effector caspase-3/7 (Figure 7).

Caspase-3/7 activity in blood cells after 24 h treatment with F-UBO shows significantly increased values than 5% P407: EA: 37.31 ± 5.88 vs. 31.69 ± 4.33, *p* < 0.05. Consequently, the blood cell’s viability was diminished considerably compared to the C2P positive control (55.06 ± 2.34 vs. C2P: 64.80 ± 3.41, *p* < 0.01, Figure 7A,D,I).

In CLS-354 tumor cells, the proapoptotic signal appreciably decreased compared with the C3UA positive control: 10.24 ± 0.76; vs. 27.02 ± 1.64, *p* < 0.01 (Figure 7B,H,J). The CLS-354 tumor cell’s viability remained on significantly higher levels, compared to the C2P and C3UA controls: 87.14 ± 1.45 vs. C2P: 85.18 ± 1.59, *p* < 0.05; C3UA: 39.25 ± 1.88, *p* < 0.01 (Figure 7B,G,H,J).

#### 3.6.3. Evaluation of Nuclear Condensation and Lysosomal Activity

Magic Red^®^ Caspase-3/7 Assay Kit [69] contains Hoechst 33,342 and acridine orange stains. Hoechst 33,342 is a cell-permeant nuclear stain; when it is linked to double chain DNA, it emits blue fluorescence, highlighting condensed nuclei in apoptotic cells [70]. Acridine orange is a chelating dye for revealing the lysosomal activity [71]. Both processes triggered in normal blood cells and CLS-354 tumor cells after 24 h treatment with F-UBO are displayed in Figure 8.

After 24 h treatment, F-UBO determined in normal blood cells significantly higher values of nuclear shrinkage compared to positive controls: 30.32 ± 1.73 vs. C2P: 19.53 ± 2.41, and C3UA: 3.19 ± 0.30, *p* < 0.01 (Figure 8A,D,E,R). The F-UBO-induced lysosomal activity is significantly lower than C2P: 42.66 ± 1.36 vs. 53.23 ± 1.99; *p* < 0.05 (Figure 8I,L,R). It is also considerably augmented compared to C3UA: 42.66 ± 1.36 vs. 27.05 ± 1.52; *p* < 0.01 (Figure 8I,M,R).

In CLS-354 tumor cells, F-UBO induced a substantially higher nuclear shrinkage than C1 negative and C2P positive controls: 49.04 ± 4.04 vs. C1: 16.11 ± 3.11, *p* < 0.05; C2P: 20.06 ± 0.37; *p* < 0.01 (Figure 8B,F,G,S). Moreover, F-UBO recorded a strong augmentation of tumor cell’s lysosomal activity compared to all controls: 66.14 ± 2.67 vs. C1: 12.57 ± 0.92; C2P: 27.27 ± 1.37; *p* < 0.01, C3UA: 53.35 ± 2.63, *p* < 0.05 (Figure 8J,N,O,P,S).

#### 3.6.4. Cell Cycle Analysis

The effects of F-UBO on the cell cycle of normal blood cells and CLS-354 tumor cells, evidenced with PI/RNase stain, are displayed in Figure 9.

In normal blood cells, F-UBO determined cell cycle arrest in G0/G1 without appreciable differences from controls: 86.99 ± 1.49 vs. C1: 88.52 ± 0.74, C2P: 85.38 ± 4.94, C3UA: 90.05 ± 3.45, *p* ≥ 0.05 (Figure 9A,C–E,I,K). However, F-UBO blocked DNA-synthesis in normal blood cells: 0.00 ± 0.00 vs. C1: 4.76 ± 0.68, *p* < 0.01; C2P: 1.79 ± 0.36, *p* < 0.05; C3UA: 2.86 ± 0.23, *p* < 0.01 (Figure 9A,C–E,I,K).

F-UBO inhibited the CLS-354 tumor cell’s growth through cell cycle arrest in G0/G1 and G2/M, registering significant differences from both positive controls. Thus, cell cycle arrest in G0/G1 is lower (73.51 ± 1.77 vs. C2P: 83.56 ± 4.20, *p* < 0.05; C3UA: 90.05 ± 3.45, *p* < 0.01) and in G2/M is higher (15.40 ± 0.72 vs. C2P: 5.12 ± 2.19, *p* < 0.05; C3UA: 4.06 ± 1.45, *p* < 0.01) compared to C2P and C3UA (Figure 9B,G,H,J,L).

#### 3.6.5. Evaluation of Total ROS Activity

Total ROS activity evaluation measured the cellular oxidative stress induced by F-UBO in blood cells and CLS-354 tumor cells. The results are displayed in Figure 10.

F-UBO-induced oxidative stress in normal blood cells was substantially higher compared to all controls: 5600 × 10^4^ ± 500.00; vs. C1: 242.00 × 10^4^ ± 2.00, C2P: 311 × 10^4^ ± 9.64; C3UA: 846.66 × 10^4^ ± 5.77, *p* < 0.01 (Figure 10A,C).

Moreover, ROS levels in CLS-354 cells treated with F-UBO considerably increased compared to C1 and C2P: 445.00 × 10^4^ ± 8.66 vs. C1: 15.66 × 10^4^ ± 4.04; and C2P: 96.66 × 10^4^ ± 20.81; *p* < 0.01 (Figure 10B,D). However, F-UBO-induced ROS levels were significantly lower than C3UA’s: 966.66 × 10^4^ ± 57.73, *p* < 0.01 (Figure 10B,D).

#### 3.6.6. Evaluation of Cell Proliferation

The F-UBO effects on DNA synthesis in normal blood cells and CLS-354 tumor cells were also assessed by EdU incorporation, and the results are presented in Figure 11.

In normal blood cells, F-UBO blocked DNA synthesis recording considerable differences from C1 negative control: 0.00 ± 0.00 vs. 10.36 ± 1.21; *p* < 0.01 (Figure 11A,C,I,K). As a result of DNA content diminution, the cell cycle arrest in subG0/G1 phase corresponds to apoptotic cell fraction; these cells have less DNA than healthy ones due to DNA fragmentation [72]. However, F-UBO induced apoptotic cell fraction (subG0/G1) [73] had significantly lower values compared with 1% DMSO (0.84 ± 0.09 vs. 2.01 ± 0.20, *p* < 0.05) and higher ones compared to UA (0.84 ± 0.09 vs. 0.00 ± 0.00, *p* < 0.01).

In CLS-354 tumor cells, F-UBO also significantly diminished DNA synthesis compared to C1 negative control: 3.09 ± 1.60 vs. C1: 12.44 ± 2.80, *p* < 0.05 (Figure 11B,F,J,L), but F-UBO-induced cell cycle arrest in subG0/G1 was still considerably lower than that of C1: 1.77 ± 1.37 vs. 15.18 ± 2.17, *p* < 0.01 (Figure 11B,F,J,L). Moreover, P407 reduced DNA synthesis in CLS-354 tumor cells higher than F-UBO: 1.16 ± 1.07 vs. 3.09 ± 1.60, *p* < 0.05 (Figure 11B,G,J,L), but apoptotic cell fraction also remained minimal.

#### 3.6.7. Principal Component Analysis

The principal component analysis (PCA) was performed for F-UBO and controls (C1-DMSO, C2P407, and C3UA). It correlates the variable parameters determined in both cell types (normal blood cells and CLS-354 OSCC tumor cells) according to the correlation matrix and PCA-correlation circle from the Appendix A. The results are displayed in Figure 12.

The two principal components explained 83.36% of total data variance, with 51.50% attributed to the first (PC1) and 31.85% to the second (PC2). PC1 was associated with C3UA, caspase-3/7 activity in normal blood cells and CLS-354 tumor cells, and ROS levels in CLS-354 tumor cells. At the same time, PC2 was related to F-UBO bioadhesive oral films, C1DMSO, and ROS levels in normal blood cells (Figure 12).

In normal blood cells, caspase-3/7 activation shows a high positive correlation with nuclear condensation (*r* = 0.921, *p* > 0.05), a moderate one with necrosis (*r* = 0.786, *p* > 0.05), ROS levels (*r* = 0.708, *p* > 0.05), and autophagy (*r* = 0.581, *p* > 0.05), and a low one with apoptotic cell fraction (subG0/G1, *r* = 0.382, *p* > 0.05). This mechanism is highly negatively correlated with EA and LA (r = −0.835, *p* > 0.05) and moderate with a cell cycle arrest in G0/G1 phase and DNA synthesis (r = −0.741 and −0.669, *p* > 0.05). The cellular oxidative stress reported a considerable positive correlation with necrosis (*r* = 0.992, *p* < 0.05), and a low one with nuclear condensation (*r* = 0.540, *p* > 0.05).

In CLS-354 tumor cells, caspase-3/7 activation is highly positively correlated with cell cycle arrest in G0/G1 phase (*r* = 0.800, *p* > 0.05), and low with ROS level (*r* = 0.513, *p* > 0.05). However, oxidative stress (expressed as ROS level) shows a high and moderate positive correlation with the most damaging processes in OSCC cells. It displays a high correlation with late apoptosis and nuclear condensation (*r* = 0.812 and 0.802, *p* > 0.05), and a moderate one with early apoptosis and autophagy (*r* = 0.739 and 0.733, *p* > 0.05).

Data analysis shows that F-UBO acts on CLS-354 cells, inducing the highest levels of nuclear condensation and autophagy compared to all controls. Moreover, nuclear condensation and autophagy triggered by F-UBO display higher levels in OSCC cells than in normal blood ones. F-UBO also causes the most elevated oxidative stress in normal blood cells compared to controls. However, F-UBO significantly diminishes the apoptotic cell fraction (subG0/G1) and autophagy (A) and slowly decreases the cell cycle arrest in G0/G1, triggered in normal cells by 1% DMSO.

Usnic acid, the main secondary metabolite of *U. barbata* lichens, induced the highest oxidative stress and caspase-3/7 activation in OSCC cells, leading to the most substantial cellular apoptosis. It highlights a significant protective effect on normal blood cells, appreciably diminishing caspase-3/7 activation, nuclear condensation, and autophagy determined by 1% DMSO, thus reducing apoptotic cell fraction (subG0/G1).

In the present study, 5% P407, the emulsifier used in the F-UBO formulation, was selected as a positive control. It significantly acts on OSCC cells, determining cellular apoptosis by triggering all mechanisms that lead to cancer cell death. Moreover, it induces the highest DNA synthesis compared to F-UBO and controls. In normal blood cells, it generates oxidative stress and caspase-3/7 activation after 24 h of treatment, but the cell viability is not significantly affected.

By correlating and interpreting these data, the places of F-UBO and controls (C1DMSO, C2P407, and C3UA) in the PCA-correlation biplot (Figure 12) were justified, evidencing the corresponding processes triggered in CLS-354 cancer cells and normal blood cells.

## 4. Discussion

The previous UBO analysis measured the heavy metals content, quantified the active constituents (UA content = 0.915 ± 0.018 mg/g UBO), explored the antioxidant, cytotoxic, and rheological properties, and then proved its suitability for pharmaceutical formulation [26].

The use of HPMC in a 15% aqueous dispersion ensured the suitable film toughness, while 5% PEG 400 provided an elegant, glossy, smooth appearance and high flexibility. The film’s homogeneity proved that the active ingredients were adequately incorporated into the polymer matrix, emulsifying the oil extract. All formulations led to a thin and uniform film, suitable characteristics for bioadhesive performance, and a comfortable administration.

The low variation in weight and thickness guarantees the efficiency of the formulation and applied method and provides a certain uniformity of content. The results obtained for film thickness agree with other developed studies on HPMC films [74]. Both types of formulations proved to have weight and thickness suitable for application to the oral mucosa [36,74,75].

Regarding the mechanical properties of both bioadhesive oral films (F-UBO and R), the differences between formulations are not substantial because they contain identical amounts of HPMC and PEG 400.

Thus, the film’s flexibility is mandatory for easy handling and administration. It is induced by the plasticizer used in the formulation and the film-forming polymer [76]. Semalty M. et al. [77] proved that mixing HPMC with PEG 400 in 30% of the weight of the polymer leads to low folding endurance and that the optimal plasticizer is PEG, used in low concentrations. In the present study, using PEG 400 in a 5% concentration led to the excellent flexibility of both film types. The plasticizer reduces the film’s rigidity by decreasing the intermolecular forces [78]. Still, it was proven that high amounts of plasticizer might diminish the film adhesive properties by over-hydration [79].

F-UBO contains the active ingredient emulsified in the polymer matrix, leading to its physical interruption and a suitable elasticity. The disruption of polymer molecular chains induces higher chain mobility, augmenting flexibility and diminishing rigidity. Maher et al. [80] proved the influence of the polymer type (including HPMC) on the film’s tensile strength. It was also confirmed that the tensile strength increases with the film-forming polymer concentration. It was observed that 15% HPMC water dispersion conducts to the development of a strong matrix with a sufficiently dense network. The obtained results show that even if the film-forming agent and the plasticizer have the primary influence on the film’s mechanical attributes, other factors, such as the active ingredient nature or concentration and its dispersion type, affect the bioadhesive film’s strength. The data obtained in this study show that F-UBO’s elongation and tensile strength are adequate to resist stress during handling [81].

The moisture content ensures the suitable film’s mechanical properties. It influences the film’s friability; however, both formulations (F-UBO and R) display good resistance. The moisture can be due to the solvent system used in the formulation or to the ingredients’ hygroscopic properties, especially the plasticizer ones [82]. PEG 400 presents high hygroscopicity due to its hydrophilic hydroxyl groups interacting with water [83], providing more sites for interactions and leading to moisture retention in the films. HPMC also has hydrophilic hydroxypropyl substituents, but contains hydrophobic methoxyl groups and does not maintain excessive moisture [84]. Bioadhesive oral films must have a moderate moisture content to ensure their elasticity and protection from being brittle, dry, and easy to break [85], and F-UBO and R show suitable humidity.

Each ingredient influences the film’s pH value in the formulation. The purpose is to properly select the components to obtain bioadhesive films with a surface pH close to the buccal one. F-UBO shows an approximately neutral pH on the surface, close to the oral cavity, ensuring good tolerability with no possible irritation of the buccal mucosa. In this work, both films have similar pH values, proving that the active ingredients do not modify the pH of the matrix system.

The formulation’s disintegration time strongly depends on the polymer matrix. Shen et al. [86] demonstrated that the film’s disintegration time rises directly proportional to HPMC concentration. The results show that F-UBO’s rapid disintegration allows a fast release of active ingredients, suitable for in vitro studies.

The film’s swelling properties are essential for bioadhesion [87] and highly depend on water diffusivity into the polymer [88]. The residence time [41,42] also depends on the film-forming polymer and the plasticizer’s retention properties, being highly controlled by the ratio between them [89]. The disturbance of the polymer chains by including active ingredients in the matrix decreases the water content [90] and considerably influences the bioadhesive behavior. Adhesion is enhanced with increasing hydration until it reaches an optimum point. Overhydration causes the breaking of the polymer/tissue interface, thus decreasing the bioadhesive force. The values registered for ex vivo residence time are strongly related to the film’s in vivo bioadhesive performance, and results were satisfactory for F-UBO.

Generally, the differences between UBO-loaded bioadhesive oral films and references are not statistically significant, proving that UBO does not considerably influence these previously mentioned properties.

On the other hand, P407, the emulsifier from the F-UBO formulation, is most known for its use as a surfactant in various oral hygiene products (dentifrices, mouthwashes, breath fresheners) in a concentration range of 0.3–20% [91]. Furthermore, it is particularly interesting in clinical use for surgical application due to its thermoreversible gelation and bactericidal effects on *S. aureus* [92]. Veyries et al. [93] revealed the potential of P407 for inhibiting the attachment of *S. aureus* and *S. epidermidis* and increasing their susceptibility to antibiotics once they are adherent. This study results show the significant antifungal potential of 5% P407 in water against *C. albicans* and *C. parapsilosis.*

Teanpaisan et al. [94] proposed a mixture of P407 and *Artocarpus lakoocha* (*Moraceae*) for endodontic treatment, proving its antibacterial activity against *E. fecalis.* Recently, another study proved antifungal activity of a hydroethanolic extract from *Astronium urundeuva* leaves loaded into a nanostructured lipid system with 0.5% P407^®^ against *C. albicans* and *C. glabrata* [95]. Previous studies [96,97] included the most common oral cavity pathogens responsible for various opportunistic infections in immunocompromised patients, *S. aureus*, *P. aeruginosa*, and *C. albicans.* Our F-UBO oral biofilms revealed a dose-dependent inhibitory activity against *S. aureus*, *P. aeruginosa*, *C. albicans*, and *C. parapsilosis.*

The BSL cytotoxicity assay is one of the most known methods, using *Artemia* sp.—*A. salina* [26] and *A. franciscana* [98]. Together with phytotoxicity assay on *Triticum aestivum* L.—which evaluates the *Triticum* radicle growth [99], the BSL assay is considered a rapid, simple, low-cost, and effective test to estimate various natural products’ safety for human use [100]. Okumu et al. [49] recently used *A. salina* as an animal model for the preliminary evaluation of snake-venom-induced toxicity. Nazir et al. [101] proved that the BSL assay is a significant antitumor prescreening in anticancer drug discovery. Rajabi et al. [102] described *Artemia salina* as a model organism in the toxicity assessment of nanoparticles. The present study used the BSL assay to evaluate the F-UBO cytotoxic potential, and the obtained results could be extrapolated to those on tumor cells.

Previous studies regarding the various poloxamer types’ cytotoxicity on tumor cells reported a dose-dependent action. Thus, a concentration of 30 ± 10 mg/mL of poloxamer 188 inhibited 50% of HeLa (cervical cancer) cell growth, whereas a dose 10 times lower (2–5 mg/mL) was necessary to inhibit 50% of B-16 (mouse melanoma) cell growth [91]. In this study, 50 mg/mL P407 reduced the viability of CLS-354 (oral cancer) cells to 72.51 ± 2.51% after 24 h incubation. F-UBO dispersion led to 315 µg/mL UBO and 3.15 mg/mL P407; UBO penetrated rapidly through the cell wall in emulsified form, then the onset of apoptotic processes after 24 h of treatment could be explained. The F-UBO complex composition suggests a synergy between *U. barbata* secondary metabolites with pharmacological properties [103,104], canola oil’s bioactive constituents [105], and P407 [106,107].

## 5. Conclusions

This study evaluated the pharmacological potential of the *U. barbata* (L.) Weber ex F.H. Wigg extract in canola oil (UBO) as an oral pharmaceutical formulation.

The UBO-loaded bioadhesive oral films were manufactured using P407, HPMC, and PEG 400 for their formulation. F-UBO suitability for topical administration on buccal mucosa was confirmed through complex physicochemical and pharmacotechnical analyses.

F-UBO antimicrobial and anticancer properties were investigated using P407 as a positive control. Data obtained revealed F-UBO and P407 dose-dependent inhibitory activity against the most common bacterial and fungal pathogens implied in immunosuppressed patients’ oral infections. Moreover, they highlighted in vitro antitumor effects on oral epithelial squamous cell carcinoma (CLS-354 cell line).

The present research suggests that bioadhesive oral films with *U. barbata* extract in canola oil can be considered a phytotherapeutic formulation with potential applications against oral cavity infections and neoplasia. In vivo and clinical studies could be further steps in F-UBO analysis to confirm their medical benefits.

## Figures and Tables

**Figure 1 antioxidants-11-01601-f001:**
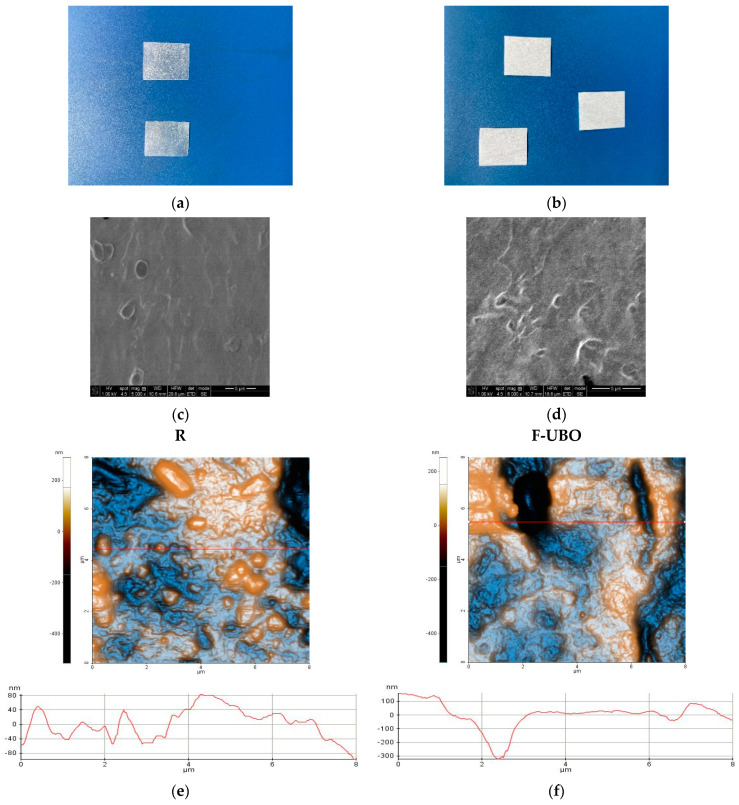
Bioadhesive oral films aspect (**a**,**b**): reference (**a**) and F-UBO (**b**). SEM images (**c**,**d**) of bioadhesive oral films: R (**c**) and F-UBO (**d**). The 2D-AFM images (**e**,**f**) of bioadhesive oral films in “enhanced contrast view” mode, at the scale of (8 × 8) µm^2^, together with representative line scans for R (**e**) and F-UBO (**f**). R—reference (a film without UBO); F-UBO—UBO-loaded bioadhesive oral film; UBO—*U. barbata* extract in canola oil.

**Figure 2 antioxidants-11-01601-f002:**
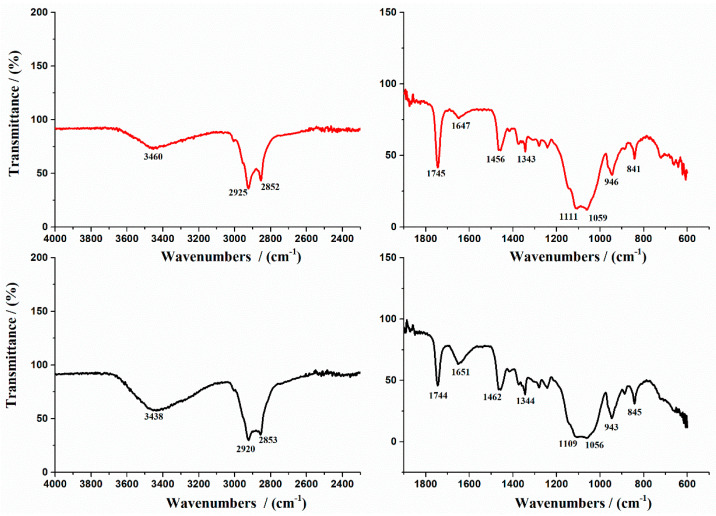
FTIR Spectra of bioadhesive oral films: R is shown with a red line and F-UBO with a black line. R—reference (a film without UBO); F-UBO—UBO-loaded bioadhesive oral film; UBO—*U. barbata* extract in canola oil.

**Figure 3 antioxidants-11-01601-f003:**
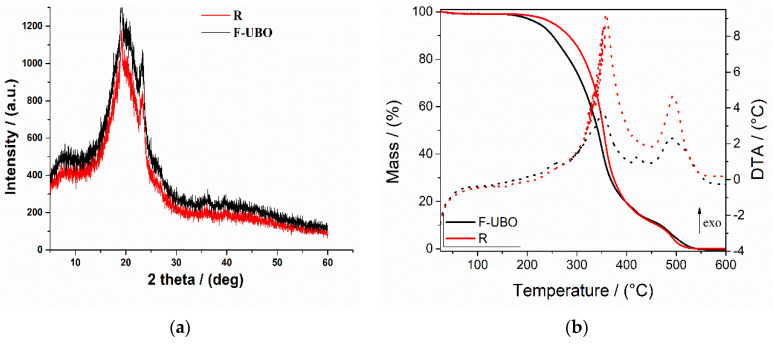
(**a**) X-Ray diffractograms of bioadhesive oral films (R and F-UBO); (**b**) thermogravimetric analysis coupled with differential thermal analyses of R and F-UBO. R—reference (a film without UBO); F-UBO—UBO-loaded bioadhesive oral film; UBO—*U. barbata* extract in canola oil.

**Figure 4 antioxidants-11-01601-f004:**
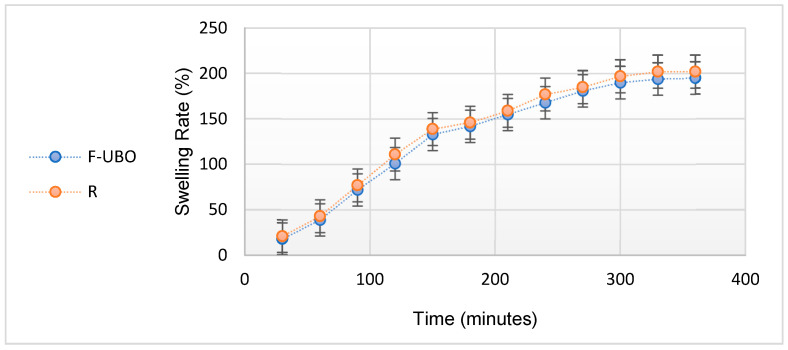
Swelling rate (%) over 6 h of F-UBO vs. R; F-UBO—UBO-loaded bioadhesive oral film; UBO—*U. barbata* extract in canola oil; R—reference (a film without UBO).

**Figure 5 antioxidants-11-01601-f005:**
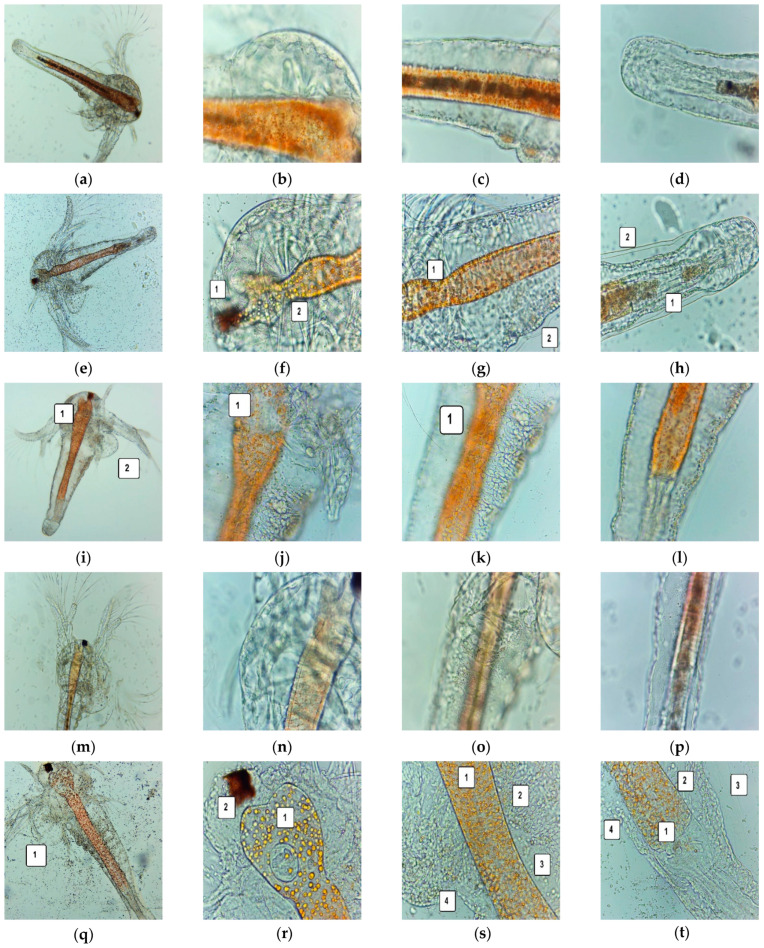
*A. salina* larvae after 24 and 48 h—microscopic images at 100× (**a**,**e**,**i**,**m**,**r**) and 400× (**b**–**d**,**f**–**h**,**j**–**l**,**n**–**p**,**s**–**u**). After 24 h: blank (**a**–**d**), F-UBO (**e**–**h**), 5% P407 (**i**–**l**); after 48 h: blank (**m**–**p**), F-UBO (**r**–**u**), 5% P407 (**v**–**x**). The following changes were observed compared to blank: (**f**) penetration of emulsified lipids into tissues (1); (**g**) penetration of emulsified lipids into tissues (1), depletion of cellular structures (2); (**h**) penetration of emulsified lipids into tissues (1), detachment of the cuticle from the terminal portion of the digestive tract (2); (**i**) increasing the volume of the digestive tube in the upper part (1); growth cessation—the brine shrimp larvae have not passed into the next stage of development (2); (**j**) different changes in the upper part of the digestive tube (1); (**k**) narrowing of the digestive tract in the lower half; (**q**) dead larvae (1); (**r**) massive penetration of small particles of emulsified lipids, into tissues (1), tissue damage (2); (**s**) blocked digestive tract due to accumulated lipids (1), accumulation of lipids in tissues (2), cell damage with large intercellular spaces (3), tissue destruction (4); (**t**) digestive tract blocked by accumulated lipids (1), cell damage with large intercellular spaces (2), massive detachment of the cuticle from larval tissues (3), tissue destruction (4); (**u**) intensifying the increase in the digestive tract’s volume (1) and growth cessation (2); (**v**,**w**) intensifying the increase in the digestive tract’s volume (1) and penetration of emulsified lipids into tissues (2); (**x**) penetration of emulsified lipids into tissues (1). *A. salina* larvae after 48 h exposure to F-UBO, stained with acridine orange 400× (**A**,**E**) and 200× (**B**–**D**,**F**): blank (**A**–**D**) and F-UBO (**E**,**F**). The red fluorescence shows intracellular lysosomes activated in cell death processes (**F**).

**Figure 6 antioxidants-11-01601-f006:**
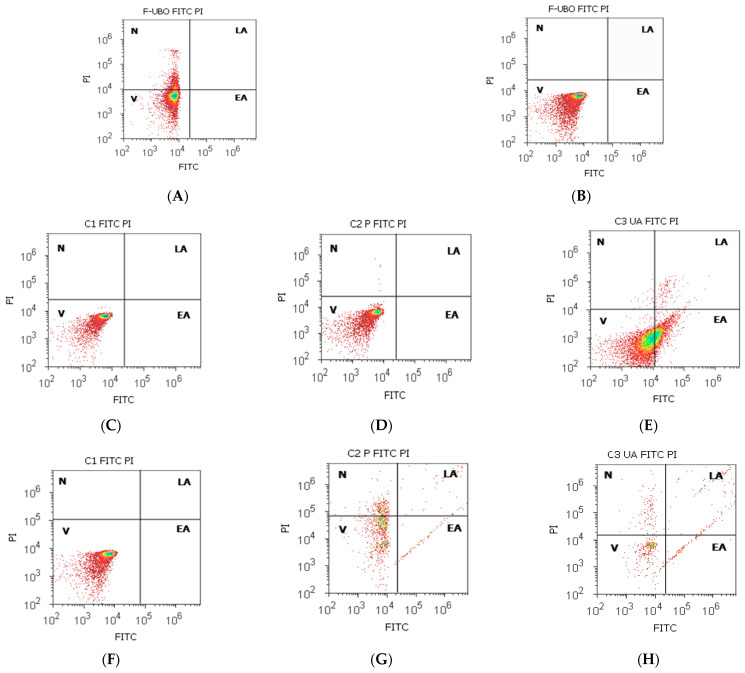
Cell apoptosis models after 24 h treatment with F-UBO in normal blood cells (**A**,**C**–**E**) and CLS-354 tumor cells (**B**,**F**–**H**). Annexin V-FITC/PI patterns of F-UBO (**A**,**B**); 1% DMSO negative control (**C**,**F**); 5% poloxamer 407 positive control (**D**,**G**); 125 µg/mL UA positive control (**E**,**H**). Statistical analysis of cell apoptosis (**I**,**J**) in normal blood cells (**I**) and CLS-354 tumor cells. * *p* < 0.05 and ** *p* < 0.01 are significant statistical differences between controls and sample (F-UBO) made by paired samples *t*-test. V—viability; EA—early apoptosis; F-UBO—bioadhesive oral films loaded with *U. barbata* extract in canola oil; C1—negative control with 1% dimethyl sulfoxide; C2P—positive control with 5% poloxamer 407; C3UA—positive control with 125 µg/mL usnic acid.

**Figure 7 antioxidants-11-01601-f007:**
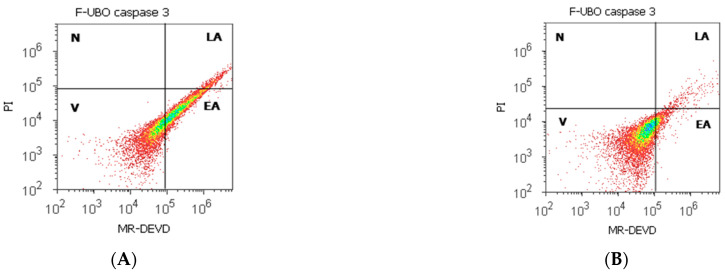
Caspase-3/7 activity after 24 h treatment with F-UBO in normal blood cells (**A**,**C**–**E**) and CLS-354 tumor cells (**B**,**F**–**H**); MR-DEVD patterns of F-UBO (**A**,**B**), 1% DMSO negative control (**C**,**F**); 5% poloxamer 407 positive control (**D**,**G**); 125 µg/mL UA positive control (**E**,**H**). Statistical analysis of caspase-3/7 activity (**I**,**J**) in normal blood cells (**I**) and CLS-354 tumor cells (**J**). * *p* < 0.05 and ** *p* < 0.01 are significant statistical differences between controls and sample (F-UBO) made by paired samples *t*-test. V—viability; EA—early apoptosis; LA—late apoptosis; N—necrosis; F-UBO—bioadhesive oral films loaded with *U. barbata* extract in canola oil; C1—negative control with 1% dimethyl sulfoxide; C2P—positive control with 5% poloxamer 407; C3UA—positive control with 125 µg/mL usnic acid.

**Figure 8 antioxidants-11-01601-f008:**
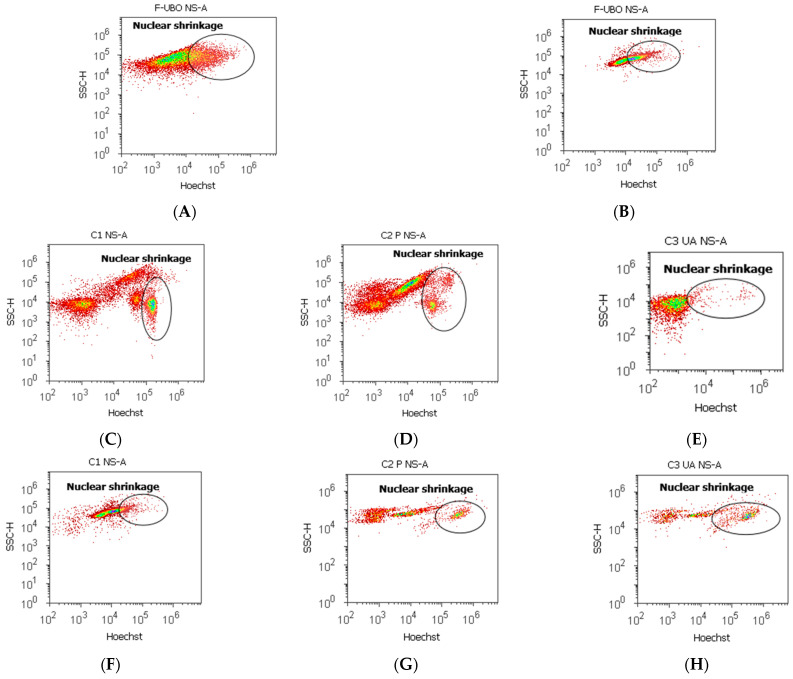
Nuclear shrinkage (**A**–**H**) and lysosomal activity (**I**–**P**) after 24 h treatment with F-UBO in normal blood cells (**A**,**C**–**E**,**I**,**K**–**M**) and CLS-354 tumor cells (**B**,**F**–**H**,**J**,**N**–**P**). Hoechst (**A**–**H**) and acridine orange (**I**–**P**) patterns of F-UBO (**A**,**B**,**I**,**J**); 1% DMSO negative control (**C**,**F**,**K**,**N**); 5% P407 positive control (**D**,**G**,**L**,**O**); 125 µg/mL UA positive control (**E**,**H**,**M**,**P**); Statistical analysis of nuclear shrinkage and autophagy (**R**,**S**) in normal blood cells (**R**) and CLS-354 tumor cells (**S**).* *p* < 0.05 and ** *p* < 0.01 represent significant statistical differences between controls and sample (F-UBO) made by paired samples *t*-test; NS—nuclear shrinkage; A—autophagy; F-UBO—bioadhesive oral films loaded with *U. barbata* extract in canola oil; C1—negative control with 1% dimethyl sulfoxide; C2P—positive control with 5% poloxamer 407; C3UA—positive control with 125 µg/mL usnic acid.

**Figure 9 antioxidants-11-01601-f009:**
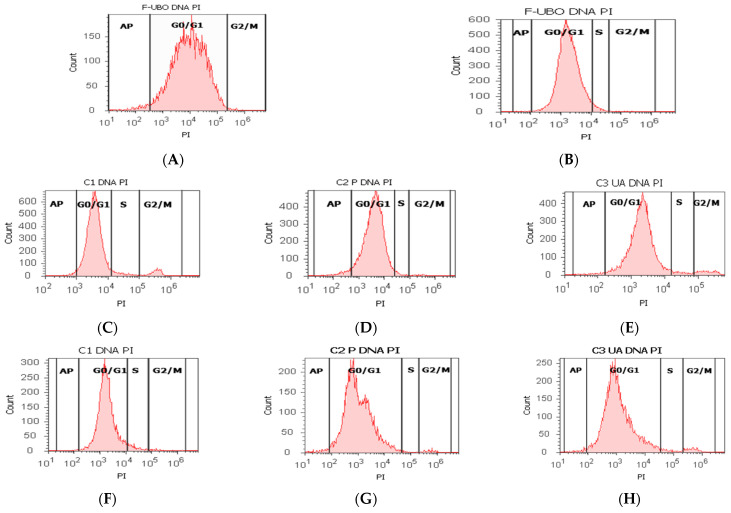
Cell cycle analysis after 24 h treatment with F-UBO in normal blood cells (**A**,**C**–**E**) and CLS-354 tumor cells (**B**,**F**–**H**). PI/RNase patterns of F-UBO (**A**,**B**); 1% DMSO negative control (**C**,**F**); 5% P407 positive control (**D**,**G**); 125 µg/mL UA positive control (**E**,**H**); F-UBO and controls extrapolated on PI axis (**I**,**J**); Statistical analysis of G0/G1, DNA synthesis (**S**), and G2/M phases of the cell cycle (**K**,**L**) in normal blood cells (**K**) and CLS-354 tumor cells (**L**). * *p* < 0.05 and ** *p* < 0.01 represent significant statistical differences between controls and sample (F-UBO) made by paired samples *t*-test; AP—apoptotic cell fraction (subG0/G1) [72]; PI—propidium iodide; S—synthesis of cell cycle phases; F-UBO—bioadhesive oral films loaded with *U. barbata* extract in canola oil; C1—negative control with 1% dimethyl sulfoxide; C2P—positive control with 5% poloxamer 407; C3UA—positive control with 125 µg/mL usnic acid.

**Figure 10 antioxidants-11-01601-f010:**
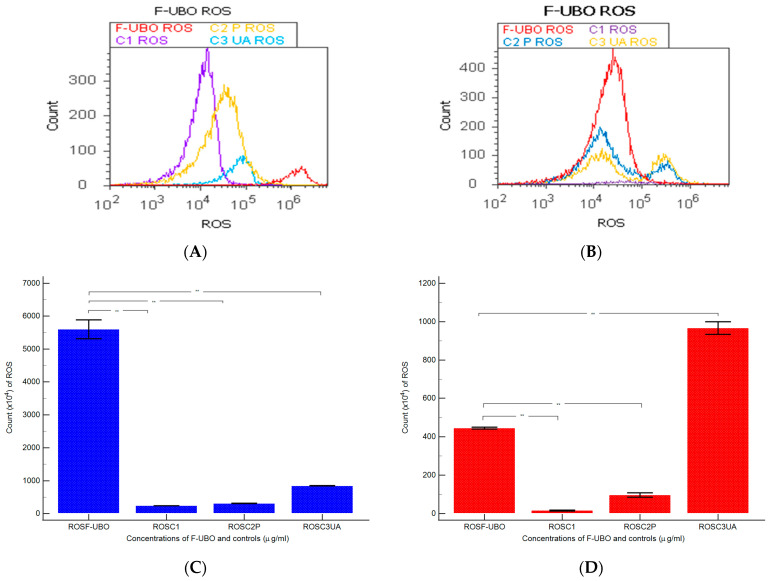
ROS levels after 24 h treatment with F-UBO in normal blood cells (**A**) and CLS-354 tumor cells (**B**) illustrated as F-UBO and controls extrapolated on the ROS axis. Statistical analysis of cellular oxidative stress (**C**,**D**) in normal blood cells (**C**) and CLS-354 tumor cells (**D**); ** *p* < 0.01 represents significant statistical differences between controls and sample (F-UBO) made by paired samples *t*-test; ROS—reactive oxygen species; F-UBO—bioadhesive oral films loaded with *U. barbata* extract in canola oil; C1—negative control with 1% dimethyl sulfoxide; C2P—positive control with 5% poloxamer 407; C3UA—positive control with 125 µg/mL usnic acid.

**Figure 11 antioxidants-11-01601-f011:**
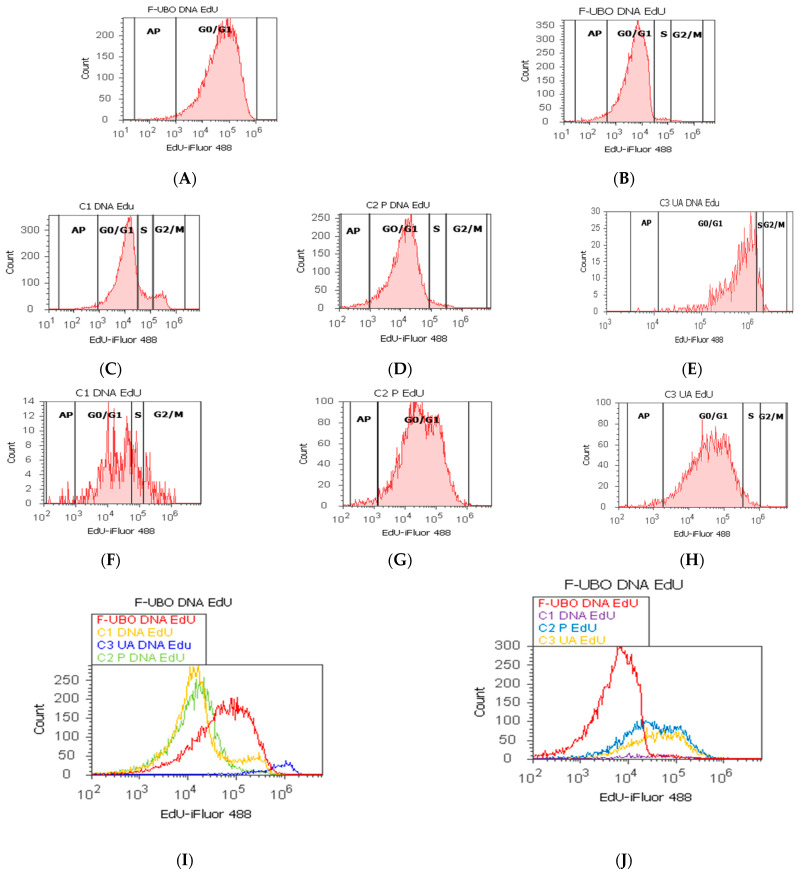
DNA synthesis (S) in normal blood cells (**A**,**C**–**E**) and CLS-354 tumor cells (**B**,**F**–**H**) after 24 h treatment with F-UBO; EdU-iFluor 488 patterns of F-UBO (**A**,**B**); 1% DMSO negative control (**C**,**F**); 5% poloxamer 407 positive control (**D**,**G**); 125 µg/mL UA positive control (**E**,**H**); F-UBO and controls extrapolated on EdU-iFluor 488 axis (**I**,**J**); Statistical analysis of DNA synthesis (**K**,**L**) in normal blood cells (**K**) and CLS-354 tumor cells (**L**); * *p* < 0.05 and ** *p* < 0.01 represent significant statistical differences between controls and sample (F-UBO) made by paired samples *t*-test. AP—apoptotic cell fraction (subG0/G1) [73]; F-UBO—bioadhesive oral films loaded with *U. barbata* extract in canola oil; C1-negative control with 1% dimethyl sulfoxide; C2P—positive control with 5% poloxamer; C3UA—positive control with 125 µg/mL usnic acid.

**Figure 12 antioxidants-11-01601-f012:**
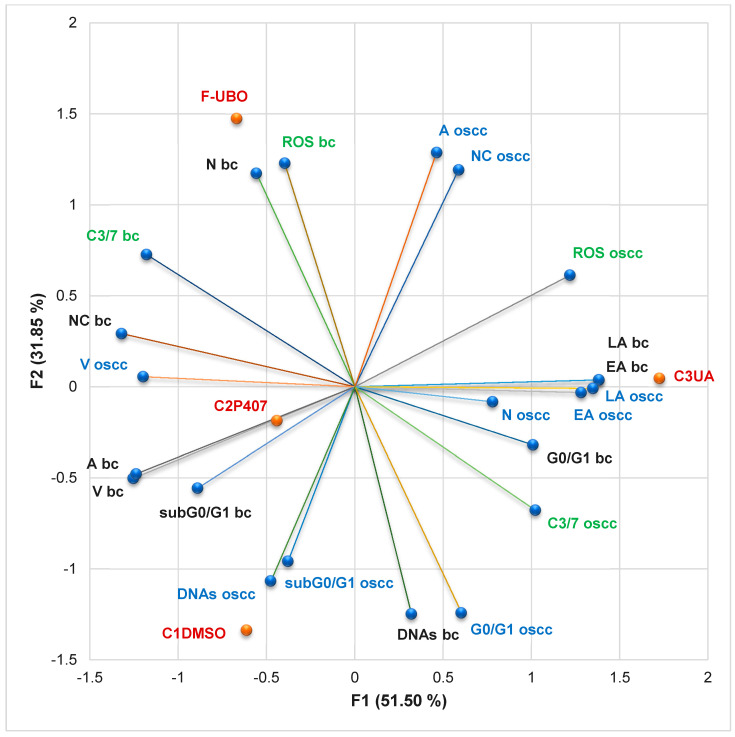
PCA-correlation biplot between mechanisms (caspase-3/7 activity and cellular oxidative stress) and processes induced by F-UBO and controls (C1DMSO, C2P407, and C2UA) in normal blood cells (bc) and CLS-354 tumor cells (oscc). V—viability; EA—early apoptosis; LA—late apoptosis; N—necrosis; NC—nuclear condensation; A—autophagy; DNAs—DNA synthesis; subG0/G1—apoptotic cell fraction; G0/G1—cell cycle arrest in G0/G1; ROS—oxidative stress; C3/7—caspase-3/7 activity.

**Table 1 antioxidants-11-01601-t001:** UBO-loaded bioadhesive oral film (F-UBO) vs. reference (R).

Ingredients	F-UBO	R
UBO	12.50	-
P407	5.00	5.00
PEG 400	5.00	5.00
HPMC 15% water dispersion (*w*/*w*)	77.8	90.00

R—reference (a film without UBO); F-UBO—UBO-loaded bioadhesive oral film; UBO—*U. barbata* extract in canola oil; PEG—polyethylene glycol; HPMC—hydroxypropyl methylcellulose; P407—poloxamer 407.

**Table 2 antioxidants-11-01601-t002:** Thermal parameters for the bioadhesive oral films (F-UBO and R) decomposition in air.

Film	Solvent Mass Loss (%)	T (°C)/Mass Loss 1st Decomposition Step (%)	T (°C)/Mass Loss 2nd Decomposition Step (%)
F-UBO	0.8	348.2 °C/87.3	488.8 °C/11.9
R	0.9	358.2 °C/86.4	A shoulder at 420 °C 495.3 °C/12.7

R—reference (films without UBO); F-UBO—UBO-loaded bioadhesive oral film; UBO—*U. barbata* extract in canola oil.

**Table 3 antioxidants-11-01601-t003:** Pharmacotechnical [65] characterization of bioadhesive oral films (F-UBO and R).

Pharmacotechnical Parameter * [66]	F-UBO	R
Weight uniformity (mg)	63 ± 1.79	62 ± 3.27
Thickness (mm)	0.069 ± 0.006	0.065 ± 0.004
Folding endurance value	>300	>300
Tensile strength (kg/mm^2^)	2.17 ± 0.49	2.36 ± 0.98
Elongation %	56.33 ± 0.92	52.16 ± 1.22
Moisture content % (*w*/*w*)	8.11 ± 0.78	8.42 ± 0.69
pH	6.97 ± 0.01	7.02 ± 0.04
Disintegration time (seconds)	124 ± 3.67	127 ± 4.81
Swelling rate (% after 6 h)	195 ± 5.24	202 ± 5.68
Ex vivo bioadhesion time (minutes)	86 ± 4.12	91 ± 3.79

* Expressed as mean value ± SD; R—reference (films without UBO); F-UBO—UBO-loaded bioadhesive oral film; UBO—*U. barbata* extract in canola oil; SD—standard deviation.

**Table 4 antioxidants-11-01601-t004:** Initial concentrations and microdilutions for standard antibacterial and antifungal drugs, positive control, and sample.

Micro- Dilution	CTR (mg/mL)		TRF (mg/mL)	P407 (mg/mL)	F-UBO (mg/mL)
	30.230 ± 0.630	122.330 ± 0.850	10.050 ± 0.180	50.133 ± 1.305	63.533 ± 1.955
1	1.511 ± 0.043	6.117 ± 0.042	0.500 ± 0.009	2.506 ± 0.065	3.176 ± 0.097
2	0.755 ± 0.022	4.893 ± 0.034	0.250 ± 0.004	1.253 ± 0.032	1.588 ± 0.048
3	0.377 ± 0.011	3.914 ± 0.027	0.125 ± 0.002	0.626 ± 0.016	0.794 ± 0.024
4	0.188 ± 0.005	3.131 ± 0.021	0.061 ± 0.001	0.315 ± 0.008	0.397 ± 0.012
5	0.094 ± 0.002	2.505 ± 0.017	0.031 ± 0.001	0.157 ± 0.004	0.199 ± 0.008
6	0.047 ± 0.002	2.004 ± 0.014	0.015 ± 0.001	0.078 ± 0.002	0.100 ± 0.004
7	0.023 ± 0.001	1.603 ± 0.011	0.007 ± 0.001	0.039 ± 0.001	0.049 ± 0.002

CTR—ceftriaxone; TRF—terbinafine; P407—poloxamer 407; F-UBO—bioadhesive oral films with *U. barbata* extract in canola oil.

## Data Availability

Data are contained within the manuscript.

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
