# Peer review of "Evaluation of Usnea barbata (L.) Weber ex F.H. Wigg Extract in Canola Oil Loaded in Bioadhesive Oral Films for Potential Applications in Oral Cavity Infections and Malignancy"

_antioxidants, 2022, doi:10.3390/antiox11081601_

Round 1

Reviewer 1 Report

Authors present interesting and complex study which fits in the journal scope. 

Methodology and results are  extensively descibed . I would suggest that methods and analytical techniques, which are already described in author's previous papers no need to repeat or describe already cited (e.g. reference no.35, Gupta et al. ,line 298) , in general, both could be presented more briefly. The abstract should be shortened and structured.

Figures and tables are also ,by my opinion, too many, my recommendation is to leave the most representative figures and tables to keep focus on main results.

 Please add number of control blood donors non smokers.  

In introduction, paragraph (line 139-141) replace with paragraph lines 128-130 ( point out the aim of the study). 

Based on your earlier observations  on canola oil and known properties of lichens, whether combination of  the canola oil  and  Usnea lichens could have  better pharmacological or more potent synergistic effect than Usnea sp. alone? Please add comment.

Reviewer 2 Report

The authors develop and extensively characterise a plant extract for use as bioadhesive oral film. I have the following comments:

MAJOR

- how did the authors test the bioadhesive properties? Please clarify 

- the authors claim that F-UBO and P407 displayed "selective cytotoxicity" on cancer cells and this is a potentially important pint for cancer treatment. However, no normal/non malignant oral cell line was used, so the nature of this selectivity is dubious. What did the author mean by selective cytotoxicity? They should either re-phrase or add a normal cell line as a control. 

- Table 4: what do those number mean? It looks like the table shown the result on one individual experiment (there is no SD or mean values). In order to reduce experimental errors, the authors must ensure that the data shown are the result of (at least) 3 independent experiments. 

MINOR

- Consider shortening the title e.g. ...applications in oral cavity diseases or ...in oral infections and cancer/neoplasia/malignancy
- Abstract: the clinical discipline dealing with mouth diseases is better defined as "oral medicine" rather than oral pathology
- there are many websites cited in the manuscript e.g. https://innvista.com/health/herbs/usnea/ (accessed on 17 July 96 2022).
Should these be placed in the reference list instead?
-
CLS-354 cells, please define type and origin in the materials and methods

- Cells were cultured in DMEM High Glucose etc: were these cells grown without serum?

Round 2

Reviewer 2 Report

The authors have adequately addressed my comments and I have no further criticisms to raise.